# Gut microbiota contribute to high-altitude adaptation in tree sparrows

Tingbei Bo,[1,2] Gang Song,[1,3] Mengru Zhang,[1] Xiaoming Xu,[1,3] Jundong Duan,[1,3] Huishang She,[1,3] Yun Fang,[1,3] Wenting Li,[4] Jing Wen,[4] Jingsong Liu,[4] Dehua Wang,[5] Fumin Lei[1,3]

**ABSTRACT** The intricate relationship between gut microbiota and various physiological functions in animals has emerged as a focal point in understanding host adaptability. Unlike the native birds of the Qinghai-Tibet Plateau (QTP), the tree sparrow (*Passer montanus*) is believed to have colonized the plateau within the last few thousand years. Given the vast expanse and harsh conditions of the plateau, the role of gut microbiota in facilitating the tree sparrow's adaptation to this high-altitude habitat remains largely unexplored and holds significant scientific interest. Therefore, we employed a multidisciplinary approach combining amplicon sequencing, transcriptome analysis, and fecal microbiota transplantation (FMT) to investigate the functional role of gut microbiota in high-altitude tree sparrows across different seasons. Results indicate that the gut microbiota of tree sparrows exhibits seasonal and altitude-dependent changes, with an increase in *Lactobacillus* in winter, which may promote heat production to cope with the cold. FMT experiments confirmed that "high-altitude gut microbiota" enhances the expression of heat-related proteins (avUCP) and upregulates heat-related genes *syt1* and *chodl*. These findings suggest an adaptive strategy whereby tree sparrows utilize their gut microbiota to modulate energy metabolism, ultimately conserving energy in the resource-limited high-altitude environment.

**IMPORTANCE** This study provides one evidence that gut microbiota mediates high-altitude adaptation in tree sparrow. By integrating multi-omics and fecal transplantation in tree sparrows (*Passer montanus*)—a species invading the Qinghai-Tibet Plateau within millennia—we reveal seasonally dynamic microbial strategies critical for survival in extreme environments. These findings establish gut microbiota as a key driver of rapid altitudinal adaptation, offering new insights into how microbial functions enable vertebrate range expansion into challenging ecosystems. The mechanistic framework also informs conservation strategies for wildlife facing climate-driven habitat shifts.

**KEYWORDS** energy balance, gut microbiota, plateau, transcriptome, tree sparrow

In recent years, the complex interplay between high-altitude environments and the gut microbiota of animals has garnered significant scientific attention. It is increasingly recognized that these environments not only shape the structure and composition of gut microbiota but also that the microbiota plays a pivotal role in facilitating host adaptation to such conditions. The Qinghai-Tibet Plateau (QTP) represents a unique ecological niche characterized by pronounced hypoxia, frigid temperatures, and intense ultraviolet radiation, posing significant challenges to its resident fauna, including both vertebrates and invertebrates (1). Over millions of years, native species on the QTP have evolved intricate genetic traits enabling them to thrive in these high-altitude environments, manifesting in diverse morphological, anatomical, physiological, and biochemical adaptations. Among avian species, those adapted to high altitudes exhibit a range

**Peer Reviewer** Huan Li, Lanzhou University, Lanzhou, China

Address correspondence to Fumin Lei, leifm@ioz.ac.cn, or Dehua Wang, dehuawang@sdu.edu.cn.

Tingbei Bo, Gang Song, Mengru Zhang, and Xiaoming Xu contributed equally to this article. The order of names among these authors was agreed upon mutually by the contributors.

The authors declare no conflict of interest.

See the funding table on p. 15.

of physiological characteristics that facilitate survival under hypoxic conditions, such as enhanced ventilatory responses, increased cardiac output, and optimized aerobic metabolism in skeletal and cardiac muscles (2, 3). These adaptations have been observed in various species, including the Bar-headed goose, Andean waterfowl (4, 5), Japanese quail (6), and other passerine birds (7). These physiological traits often correlate with genetic modifications at the molecular level, particularly in genes involved in oxygen transport and utilization, such as hemoglobin and cytochrome C oxidase genes, which have undergone adaptive evolution to optimize performance in hypoxic environments (8). Similarly, the genome of the ground tit (*Pseudopodoces humilis*), a species endemic to high altitude, exhibits positive selection of many genes associated with energy metabolism, hypoxia response, and bone development (9).

With the advent of high-throughput sequencing technology, accumulating evidence highlights the functional significance of gut microbiota in high-altitude adaptation. During the cold season, the abundance of *Bacteroidetes* in Tibetan sheep (*Ovis aries*) is higher, while that of *Firmicutes* is lower (10). Similar results regarding changes associated with microbial communities were observed in wild plateau pika (*Ochotona curzoniae*) and yaks (*Bos mutus*) (11, 12). *Bacteroidetes* play an important role in the degradation of carbohydrates and proteins, indicating that, in cold seasons, when food resources are scarce, *Bacteroidetes* in the host may help to more widely utilize polysaccharides and fats (10, 13). In addition to seasonality, some population-based studies have also reported the association between spatial variability (including altitude and latitude) and genetically modified components (14). Li et al. found that compared to low altitude, pika at high altitude exhibited higher microbial diversity and cellulase activity, indicating that changes in altitude played a role in the formation of the gut microbiota structure (15). Previous studies have shown that the stability of bird microbiomes is relatively low (16, 17), and the extensibility to environmental and dietary changes is high. Previously, through the analysis of the gut microbiota of seven bird species in the wetlands of northern Tibet, we found that altitude is the main factor affecting the proportion of key bacteria (18). However, further research is needed to investigate the impact of the interaction between altitude and season on the gut microbiota of birds.

The tree sparrow exhibits a broad distribution across the Eurasian landmass. Historical evidence suggests that approximately 3,000 years ago, this species colonized the QTP, ascending to elevations exceeding 2,000 m in tandem with the expansion of human agricultural practices (19). Concurrently, highland populations of the tree sparrow have developed adaptive phenotypic modifications, including increased body size and enhanced muscle fiber thickness, distinguishing them from their low-altitude counterparts. The high-altitude phenotype demonstrates an enhanced capacity for metabolic heat production in the pectoral muscles and upregulates relevant genes, facilitating acclimatization to the stringent environmental conditions (20). Complementary studies have shown that this species modulates insulin sensitivity and enhances glucose metabolism to adapt to the hypoxic high-altitude environment (8, 21). A population genomic comparison between high- and low-altitude tree sparrows has revealed that multiple genetic loci associated with altitude-adaptive traits undergo simultaneous frequency shifts across the genome, expediting the transition of muscular phenotypic traits (19). While significant progress has been made in deciphering the genomic mechanisms underlying high-altitude adaptation, the role of gut microbiota remains relatively underexplored. We speculate that the gut microbiota may play a crucial role in the survival and maintenance of tree sparrows on QTP, and high-altitude microbiota may promote the expression of thermogenic genes and the acquisition of food energy. In order to elucidate the functional impact of the gut microbiome of high-altitude tree sparrows, this study conducted a fecal microbiome transplantation experiment. Through a comprehensive approach that includes metagenomics, transcriptomics, and energy metabolism indicators, the aim is to elucidate the strategy of tree sparrows adapting to high-altitude environments through gut microbiome regulation.

## MATERIALS AND METHODS

### Experiment 1: Tree sparrow fecal sample collection

Tree sparrow (*Passer montanus*) samples were collected from the high-altitude Gahai Zecha National Nature Reserve (102.34°E, 34.24°N, 3484 m) in Gansu Province and the low-altitude Wangjiayuan (116.04°E, 40.17°N, 161 m) in Beijing during both winter (December–January 2021) and summer (June–July 2022). The climate difference between the two places is significant. In the Gahai Zecha National Nature Reserve, the daily average temperature in summer is 12.5°C. In winter, it is −6.5°C, and at night, it can reach as low as −35.4°C. Beijing Wangjiayuan has an average daily temperature of 28.5°C in summer. In winter, the average daily temperature is 0°C, and the low temperature can reach −10°C. The sparrows were divided into four groups: WH ($n$ = 8, high-altitude winter), SH ($n$ = 10, high-altitude summer), WL ($n$ = 10, low-altitude winter), and SL ($n$ = 11, low-altitude summer). Before collection, the position of the tree sparrow was observed through binoculars to determine the position of the net; a bird net was set up; the tree sparrow was captured; and its body weight, length, and other morphological indicators were measured. The tree sparrows were euthanized with anesthetics and dissected, and the intestinal tract of the tree sparrow was isolated and placed in a sterile culture dish. The intestinal contents taken 1–2 cm near the cloaca after execution were used to put it into a sterile freezer. Because bird excrement is a mixture, we used the contents of the intestine near the cloaca similar to feces. After labeling, we stored it in liquid nitrogen and transferred it back to the laboratory for storage in a −80°C freezer. Therefore, the processing method and time for each bird were guaranteed to be the same to avoid errors. All procedures were approved by the Animal Care and Use Committee of the Institute of Zoology, Chinese Academy of Sciences (IOZ-IACUC-2022-114).

### Experiment 2: Fecal microbiota transplantation (FMT)

Zebra finches (*Taeniopygia guttata*) were selected as the recipient organism for microbiota transplantation due to their similar size (approximately 10 cm) and omnivorous diet, which is primarily composed of seeds, fruits, and insects, conferring robust adaptability. Twenty-five 12-week-old male zebra finches were obtained from a local breeder (Beijing, China) to ensure uniform environmental feeding conditions. After arriving at the Institute of Zoology, Chinese Academy of Sciences, these birds were kept in the animal center and had unrestricted access to food and water under a 12 h light/dark cycle (0630 h/1830 h) and a temperature of 23 ± 1°C. Measures have been taken to minimize noise as much as possible to prevent external interference. Prior to the experiment, zebra finches were housed in groups of four per cage (60 cm × 30 cm × 40 cm) and allowed to adapt to the environment for 1 week (22, 23). After 1 week, the birds were randomly divided into five groups, with five birds in each group, to ensure that the average weight of each group was equal. The five groups ($n$ = 5) were as follows: Con (gavaged with phosphate-buffered saline), WH-FMT (FMT with high-altitude winter sparrow stools), SH-FMT (FMT with high-altitude summer sparrow stools), WL-FMT (FMT with low-altitude winter sparrow stools), and SL-FMT (FMT with low-altitude summer sparrow stools). The donor bacteria were collected from the gut content of wild tree sparrows in Experiment 1, which were collected in PBS and kept frozen, and the content of all tree sparrows was mixed and used in each group. The contents (100 mg) from the donors were suspended in sterile 0.9% saline (2 mL) and centrifuged to get the bacterial suspension (24). Oral gavage administration was 50 uL per day. FMT was administered daily for 2 weeks with daily weight measurement. After the domestication experiment, the finches were euthanized under anesthesia to collect the following tissue samples: carcass, liver, stomach, brain, pectoral, and gut. We took the contents of the intestine near the cloaca (1–2 cm), similar to feces. All samples were frozen at −80°C.

## Extraction and amplification of bacterial DNA for sequencing

DNA was extracted from the fecal samples collected from the wild tree sparrows and zebra finches using the QIAamp Power Fecal DNA Kit (Qiagen, Germany) following the manufacturer's instructions. The V3–V4 region of the 16S rDNA of the bacteria was amplified by PCR using primers 338F (5′-ACTCCTACGGGAGGCAGCAG-3′) and 806R (5′-GGACTACHVGGGTWTCTAAT-3′). Paired-end libraries with 150 bp read length were constructed and sequenced using Illumina NovaSeq 6000 (Illumina, Inc., San Diego). We used FLASH (Version 1.2.11) software to splice the samples' reads to obtain Raw Tags. Then, fastp (Version 0.20.0) software was used to process Raw Tags to obtain high-quality Clean Tags. Finally, we used Usearch software to compare Clean Tags with the database to detect chimeras and remove them, and then the chimera sequences were removed to obtain the Effective Tags. For the Effective Tags obtained previously, denoise was performed with the DADA2 or deblur module in the QIIME2 software (Version QIIME2-202006) to obtain initial amplicon sequence variants (ASVs) (default: DADA2). We used the SILVA rRNA database (v138.2) for the taxonomic classification of quality-filtered ASVs (https://ftp.arb-silva.de).

Species annotation was performed using QIIME2 software. To study the phylogenetic relationship of each ASV and the differences of the dominant species among different samples (groups), multiple sequence alignment was performed using QIIME2 software. The absolute abundance of ASVs was normalized using a standard sequence number corresponding to the sample with the least sequences. To find out the significantly different species at each taxonomic level (phylum, class, order, family, genus, species), we used to do MetaStat and Kruskal-Wallis test analysis in R (Version 3.5.3). Sequence counts were rarefied to 10,000 reads per sample using QIIME2's feature-table rarefy command to eliminate the sampling depth bias. The Shannon index was calculated using the q2-diversity plugin in QIIME2 with default parameters (alpha diversity). Principal coordinate analysis (PCoA) based on Bray-Curtis metrics, Jaccard, weighted Unifrac distances, and unweighted Unifrac distances was used to visualize the structure of microbial community (beta diversity). Group differences in microbial community structure (visualized via PCoA) were statistically tested using permutational multivariate analysis of variance (PERMANOVA) implemented by the adonis2 function in the R package vegan (v2.6-4) with 999 permutations. Analyses were performed on Bray-Curtis distance matrices. The linear discriminant analysis (LDA) effect size (LEfSe) method was used to assess differences in microbial communities using a LDA score threshold of 3.5. Kyoto Encyclopedia of Genes and Genomes (KEGG) pathways overrepresented in the gut microbiome were predicted using PICRUSt2 based on several gene family databases (25). We used the program STAMP to conduct $t$-tests comparing the abundances of predicted functions between WH and WL groups or SH and SL groups in Experiment 1.

## Transcriptome sequencing and analysis for zebra finches

The extraction of RNA from zebra finches intestinal tissues (without content) was performed using the TianGen RNA Easy Fast Animal Tissue/Cell Total RNA Extraction Kit. A total of 25 samples of intestinal transcriptomes from Zebra Finches were sequenced using the MGI-2000 sequencing platform for eukaryotic transcriptome sequencing using PE150 (paired-end, 150 bp reads), and the data were quality-controlled and filtered after sequencing to obtain clean data for subsequent analysis. STAR + RSEM was used to calculate the gene expression levels in zebra finch intestinal tissues, and the measurement unit fragments per kilo bases per million fragments (FPKM) was used to normalize the expression levels. DESeq2 software was used to analyze the differential expression of genes, and significant differentially expressed genes (DEGs) were screened based on the conditions of log2FoldChange $> 1$ and $< -1$, and $P$-value $< 0.05$. KOBAS 3.0 was used to enrich the biological functions of the differentially expressed genes in Gene Ontology (GO) and KEGG and identify the significantly enriched functional pathways and the main biological functions performed by the differentially expressed genes.

## Histomorphology of the intestinal villi for zebra finches

In order to understand the status of intestinal villi, we conducted slice experiments. Bird intestine fragments (1 cm) were fixed in 4% paraformaldehyde at room temperature before embedding in paraffin. Tissues were sectioned at 5 μm thickness and stained with hematoxylin and eosin (H&E) staining. Tissue sections were photographed using a Nikon optical microscope (Nikon H600L). The villus length in the intestine was determined using ImageJ software.

## Cytochrome-c oxidase (COX) activity for zebra finches

The COX activity is directly associated with the ability of mitochondria to produce adenosine triphosphate (ATP), making it an important indicator for measuring cellular thermogenesis. The activities of COX in the liver and pectoral muscle were measured polarographically at 30°C using a Clark electrode. The respiration medium contained 100 mM KCl, 20 mM TES, 1 mM EGTA, 2 mM $MgCl_2$, 4 mM $KH_2PO_4$, and 60 mM BSA at pH 7.2. A 10 mL aliquot was taken from the supernatant, and 30 mL of cytochrome c (37.9 mg/mL) was added to the electrode. The activity of COX was measured in a final volume of 2 mL.

## Heat production capacity test for zebra finches

To understand the heat production capacity of bird pectoral muscles at the mitochondrial level, we measured the avUCP levels. Pectoral muscles (20–30 mg) were homogenized in RIPA buffer and cleared by centrifugation according to standard techniques. Western blots of whole-tissue lysates were probed with primary antibodies against avUCP (made by Wenzhou University). The secondary antibody used was either peroxidase-conjugated goat anti-rabbit IgG (111-035-003; Jackson). Protein markers (20351ES76; Shanghai China) were added on both sides of each gel to verify bands. The polyvinylidene fluoride (PVDF) membranes were detected by enhanced chemiluminescence (Beyotime China). Bands analyzed using Image Lab Software (Bio-Rad Laboratories) were normalized to β-actin and expressed as relative units (RU).

## Statistical analysis

Statistical analysis was conducted using the SPSS 22.0 software package and GraphPad Prism 9. In Experiment 1, Wilcoxon test and two-way ANOVA, followed by Tukey's post-hoc tests, were used to analyze Shannon and relative abundance of gut microbiota. In Experiment 2, the weight of the zebra finches was analyzed using repeated-measures ANOVA, the weight and length of other organs were tested using the one-way ANOVA with Tukey's post-hoc tests. Intergroup differences in gut microbiota were determined using the Kruskal-Wallis test. The other analysis methods for transcriptome and gut microbiota can be found in the Methods section. Pearson correlation analyses were used to determine the relationship between gut microbiota and metabolic parameters. Results were presented as means ± SEM. *$P < 0.05$, **$P < 0.01$, ***$P < 0.001$.

## RESULTS

### Differences in gut microbiota between high- and low-altitude tree sparrows

To assess the differences in gut microbiota between high- and low-altitude tree sparrows, we conducted amplicon sequencing on fecal samples collected from these birds in two distinct seasons. After quality control and filtering, a total of 1,877,443 sequences were generated, averaging 48,139 sequences per sample. Annotation of these sequences revealed a variable microbial composition among the groups. Specifically, in winter, 131 ASVs were identified in the high-altitude group, whereas 353 ASVs were annotated in the low-altitude group. Conversely, in summer, 97 ASVs were annotated for the high-altitude group, and 171 ASVs were annotated for the low-altitude group (Fig. 1A). A comparison of the Shannon index, a measure of the microbial alpha diversity, demonstrated contrasting patterns between seasons. Our results indicated that altitude

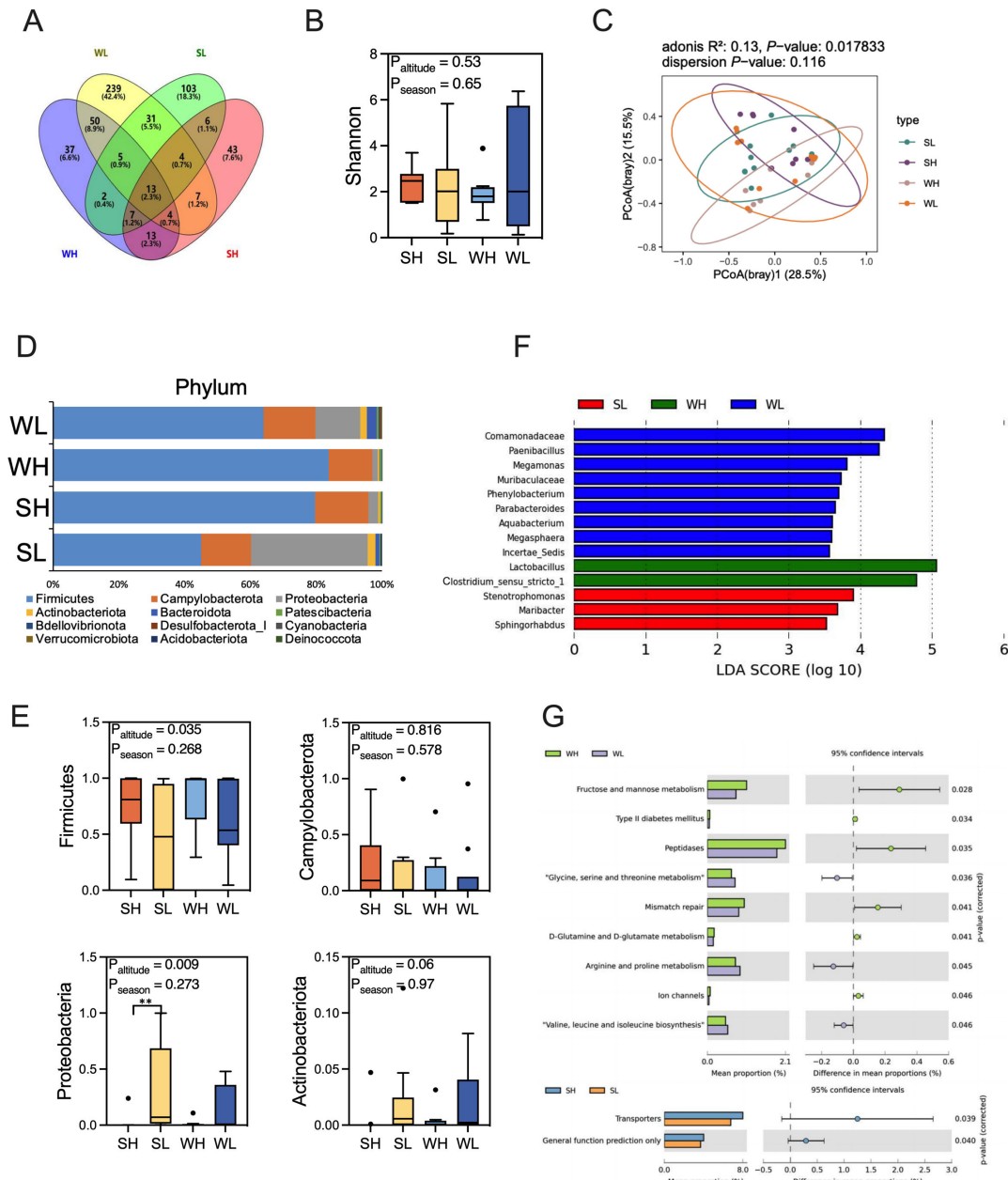

**FIG 1** Comparison of the gut microbiota of tree sparrows at high and low altitudes in winter and summer. A Venn diagram shows the number of identified amplicon sequence variants in the four groups. (B) There is no difference in alpha diversity (Shannon index, two-way ANOVA). (C) Principal-coordinate analysis (PCoA) plot of the Bray-Curtis distance (Adoins). (D) Result of the phylum level of gut microbiota in tree sparrows. (E) Relative abundance of specific phylum of four groups (Two-way two-way ANOVA). (F) Differential bacterial taxonomy selected by linear discriminant analysis effect size with an LDA > 3.5 in microbiota. (G) Functional differences in gut microbiota of tree sparrows at high and low altitudes in winter and summer (STAMP, *t* test). SH: summer high altitude; SL: summer low altitude; WH: winter high altitude; WL: winter low altitude; *P < 0.05, **P < 0.01, ***P < 0.001. n = 8–11 (WH: 8, WL: 10, SH: 10, SL: 11).

and season do not affect the alpha diversity of the gut microbiota in tree sparrows ($P_{altitude}$ = 0.53; $P_{season}$ = 0.65). In summer, the gut microbiota of high-altitude tree sparrows showed a trend toward higher Shannon index compared to low-altitude populations, though this difference was not statistically significant ($P_{altitude}$ = 0.53; $P_{season}$ = 0.65; Fig. 1B). PCoA based on Bray-Curtis distances visualized distinct clustering patterns among the four groups (PERMANOVA $P$ = 0.017, $R^2$ = 0.13, Fig. 1C). The PERMANOVA analysis showed that the diversity differences calculated based on Jaccard

and unweighted unifrac distance were significant between groups (see Table S1 at https://doi.org/10.6084/m9.figshare.29474519). Analysis of the SILVA database annotation reveals that, at the phylum level, the microbiota was predominantly composed of *Firmicutes*, *Proteobacteria*, *Campylobacterota*, and *Actinobacteriota* (Fig. 1D). In the WH group, *Firmicutes* dominated, accounting for 84.40% of the microbiota, followed by *Campylobacter* (12.39%) and *Proteobacteria* (1.93%). In contrast, the WL group showed a similar dominance of *Firmicutes* (60.41%) but with a higher proportion of *Proteobacteria* (16.93%) and *Campylobacter* (13.86%). During the summer season, the high-altitude group's gut microbiota is primarily composed of *Firmicutes* (56.62%), with *Proteobacteria* (30.12%) and *Campylobacter* (12.35%) following in abundance. However, in the low-altitude summer group, *Proteobacteria* became the most prevalent phylum (40.24%), followed by *Firmicutes* (10.07%) and *Campylobacter* (14.86%). The Wilcoxon test was conducted between the top five phyla in terms of abundance, and it was found that there were significant differences in the composition of the gut microbiota of tree sparrows at high and low altitudes in summer. *Actinobacteria* ($P = 0.025$) and *Bacteroidetes* ($P = 0.008$) were significantly lower in the high-altitude group than in the low-altitude group (see Table S2 at https://doi.org/10.6084/m9.figshare.29474519). To further investigate these differences, a two-way ANOVA test was performed on the bacterial phyla across groups. The results demonstrate that the altitude factor significantly affects the abundance of *Firmicutes* ($P_{altitude} = 0.035$; $P_{season} = 0.268$; $P_{interation} = 0.7898$) and *Proteobacteria* ($P_{altitude} = 0.009$; $P_{season} = 0.273$; $P_{interation} = 0.3194$); however, seasonal factors were not significant (Fig. 1E).

A LefSe analysis utilizing a threshold of LDA > 3.5 was conducted to investigate the gut microbiota of tree sparrows across varying altitudes and seasons. Specifically, among the winter high-altitude population, two bacterial genera emerged as dominant and significantly different: *Lactobacillus* and *Clostridium sensu stricto* 1 (Fig. 1F). In contrast, the winter low-altitude group exhibited a more diverse set of nine dominant bacterial genera with significant differences, including *Comamonadaceae*, *Paenibacillus*, *Megamonas*, *Muribaculaceae*, *Phenonobacterium*, *Parbacteroides*, *Aquabacterium*, and *Megaspaera* (Fig. 1F). However, no dominant bacterial genera with significant differences were identified in the SH group. Instead, the SL group showed three significantly abundant genera: *Stenotrophomonas*, *Maribacter*, and *Sphinghorhabdus* (Fig. 1F). The Wilcoxon test was performed on the top five genera in terms of abundance (see Table S3 at https://doi.org/10.6084/m9.figshare.29474519). The gut microbiota of tree sparrows in high-altitude areas exhibited a significant difference in the abundance of *Lactobacillus* between winter and summer, with winter showing a significantly higher abundance ($P = 0.0128$). Additionally, analysis of the predicted functional profiles of the gut microbiota revealed distinct patterns in material and energy metabolism between high- and low-altitude tree sparrows. In winter, the microbiota functions were more concentrated in fructose and mannose metabolism, peptidases, and mismatch repair, and D-glutamine and D-glutamate metabolism were higher in high-altitude sparrows. In contrast, the microbiota of low-altitude tree sparrows was more significantly associated with glycine, serine, and threonine metabolism, arginine and proline metabolism, and valine, leucine, and isoleucine biosynthesis (STAMP, *t* test, $P < 0.05$; Fig. 1G). There was no significant difference in summer.

## Effect of FMT on the physiological phenotype of zebra finches

To elucidate the effect of gut microbiota on host physiology, we performed fecal microbiota transplantation (FMT) from tree sparrows to zebra finches, as depicted in Fig. 2A. Results indicated no significant difference in body weight across five experimental groups ($P > 0.05$, Fig. 2B and C). However, the carcass weight of the WL-FMT group was higher than the control ($P = 0.045$, Fig. 2D). The stomach weight of the WL-FMT group was significantly higher than the control group, suggesting an increment in food consumption ($P = 0.024$, Fig. 2E). Furthermore, there were no differences in the liver, pectoral muscle, brain tissue weights, gut length, and weight among these five

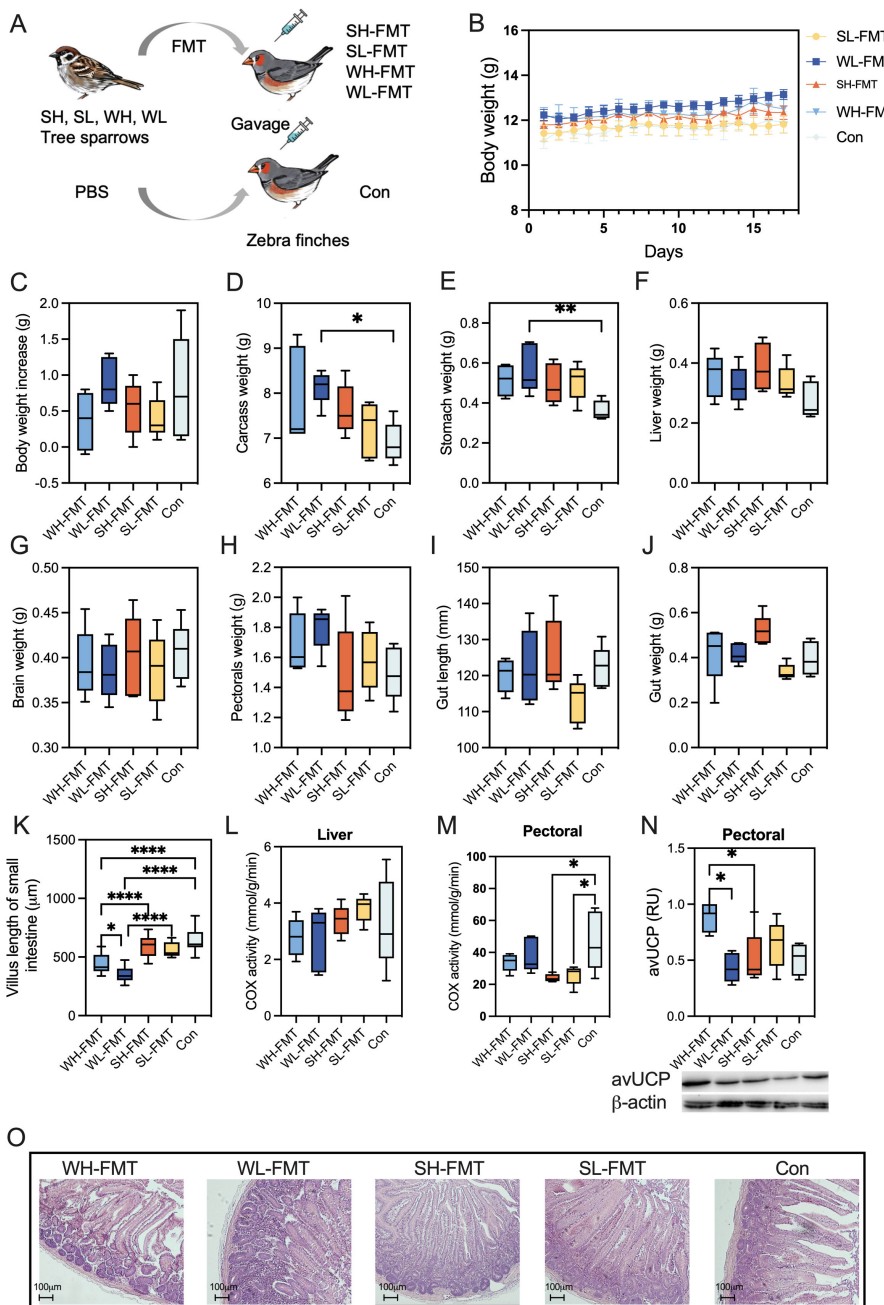

**FIG 2** Experimental results of fecal microbiota transplantation in the zebra finches. (A) Fecal microbiome transplantation pattern diagram. (B) Body weight during FMT of zebra finches (repeated ANOVA). (C) Body weight increase after FMT of zebra finches (one-way ANOVA). (D–H) Weight of the carcass, stomach, liver, brain, and pectorals of zebra finches (one-way ANOVA). (I, J) Length of the gut and weight of the gut of zebra finches (one-way ANOVA). (K) Villus length of small intestine of zebra finches (one-way Anova). (L, M) COX activity in the liver and pectorals of zebra finches. (N) avUCP (avian uncoupling protein) in the liver of zebra finches (one-way Anova). (O) Small intestine villus paraffin section HE staining of zebra finches (scale bars, 100 μm). SH-FMT: "summer high altitude microbiota" transplantation; SL-FMT: "summer low altitude microbiota" transplantation; WH-FMT: "winter high altitude microbiota" transplantation; WL-FMT: "winter low altitude microbiota" transplantation; *$P$ < 0.05, **$P$ < 0.01, ***$P$ < 0.001. $n$ = 5.

groups (Fig. 2F through J). Histological assessment employing paraffin sectioning and hematoxylin-eosin (HE) staining revealed elongated intestinal villi within the summer transplantation groups; villi in the WH-FMT group were longer than those in the WL-FMT

group ($P < 0.0001$; Fig. 2K and O). There was no difference in the COX activity in the liver ($P = 0.408$; Fig. 2L), while the COX activity in the SH-FMT and SL-FMT groups was significantly lower than that in the control group in the pectoral muscle ($P = 0.0132$; Fig. 2M). Meanwhile, the expression level of avUCP in the pectoral muscles was higher in the WH-FMT group than in the WL-FMT and SH-FMT groups ($P = 0.0159$; Fig. 2N).

## Differences in gut microbiota of zebra finch after FMT

We performed 16S rRNA gene sequencing on feces samples from zebra finches to validate the effectiveness of gut microbiota transplantation. High-throughput sequencing, quality control measures, and filtering of 25 samples yielded a total of 4,105,158 sequences, averaging 164,206 sequences per sample. Analysis of the Shannon index revealed a marginally higher gut microbiota alpha diversity in the WH-FMT and WL-FMT groups compared to others, though no statistical significance was observed (Kruskal-Wallis, $P > 0.05$; Fig. 3A, see also Table S4 at https://doi.org/10.6084/m9.figshare.29474519). For beta diversity, we constructed a PCoA plot, which clearly demonstrated significant clustering patterns among the five groups (PERMANOVA, $P = 0.000167$, $R^2 = 0.29$; Fig. 3B). At the phylum level, the gut microbiota of zebra finches was primarily composed of *Firmicutes*, *Campylobacterota*, and *Actinobacteriota* (Kruskal-Wallis, Fig. 3C). Specifically, *Campylobacter* accounted for the majority (72.2%) in the control group, whereas *Firmicutes* (67.3%) was the most prevalent in the FMT groups (Fig. 3C). Intriguingly, the *Firmicutes* abundance in the SH-FMT group was significantly elevated compared to the control (Kruskal-Wallis, $P = 0.0088$; Fig. 3D). Compared with those in the control group, *Campylobacter*, *Staphylococcus*, *Streptococcus*, *Enterococcus*, *Rothia*, *Gallibacterium*, *Vibrio*, *Pseudoalteromonas*, *Neisseria*, *Corynebacterium*, *Escherichia_Shigella*, *Riemerella*, and *Aeromonas* were all decreased in the other four groups (Fig. 3E). However, *Lactobacillus* was significantly higher in the SH-FMT group than in the control group (Kruskal-Wallis, $P = 0.0059$; Fig. 3F). *Alcaligenaceae* was significantly higher in the SH-FMT group than the other groups (Kruskal-Wallis, $P = 0.0004$; Fig. 3F). Through microbial function prediction, the results showed that the SH-FMT group had more microbiome involved in carbohydrate metabolism and lipid metabolism (LDA > 3.5; Fig. 3G). A correlation analysis based on microbial genus and physiological indexes revealed *Lactobacillus* was positively correlated with liver weight, while *Campylobacter*, *Staphylococcus*, *Rothia*, *Vibrio*, and *Aeromonas* were negatively correlated with liver weight. *Alcaligenaceae* was positively correlated with gut weight, while *Paracoccus* and *Granulicatella* were negatively correlated with villus length (Pearson, $P < 0.05$; Fig. 3H).

Transcriptomic analyses were conducted on 25 gut samples, utilizing "high-altitude tree sparrow microbiota" transplanted groups (WH-FMT vs. SH-FMT) as the experimental cohorts and "low-altitude tree sparrow microbiota" transplanted groups (WL-FMT vs. SL-FMT) as the control groups. These comprehensive assessments allowed us to rank the top 40 genes in terms of their overall abundance across the various groups (Fig. 4A). To further narrow down our focus, a differential expression analysis was performed, employing the criteria of log2 FoldChange > 1 or < −1 to identify genes with significant changes in expression. Specifically, within the gut tissue of the WH-FMT and WL-FMT groups, five genes exhibited significant differential expression, three of which were upregulated (*syt1*, *hmbox1*, *chodl*), and two were downregulated (*atl2*, *got1*) ($P < 0.05$, Fig. 4B). Conversely, in the intestinal tissue of the SH-FMT and SL-FMT groups, a total of 36 genes were found to be significantly differentially expressed, with 21 genes (like *spon2*, *lyn*, *cux1*, *tfcp2l1*, *atp2a3*, *atp6v1c2*, *cebpb*, *rbms1*, *ebf1*, *nfatc1*, *rftn1*, *foxd2*) upregulated and 15 genes downregulated (like *irs2*, *gal3st1*, *abat*, *hint2*, *gata6*) ($P < 0.05$, Fig. 4C). To gain deeper insights into the biological functions of these differentially expressed genes, we further performed a functional enrichment analysis. By selecting the pathway with the highest enrichment level and a $P$-value less than 0.05, we were able to construct functional enrichment tree plots (Fig. 4D and E). Compared to WL-FMT, WH-FMT zebra finches enriched pathways primarily related to arginine and carbon metabolism, including biosynthesis of amino acids, cysteine and methionine metabolism, arginine

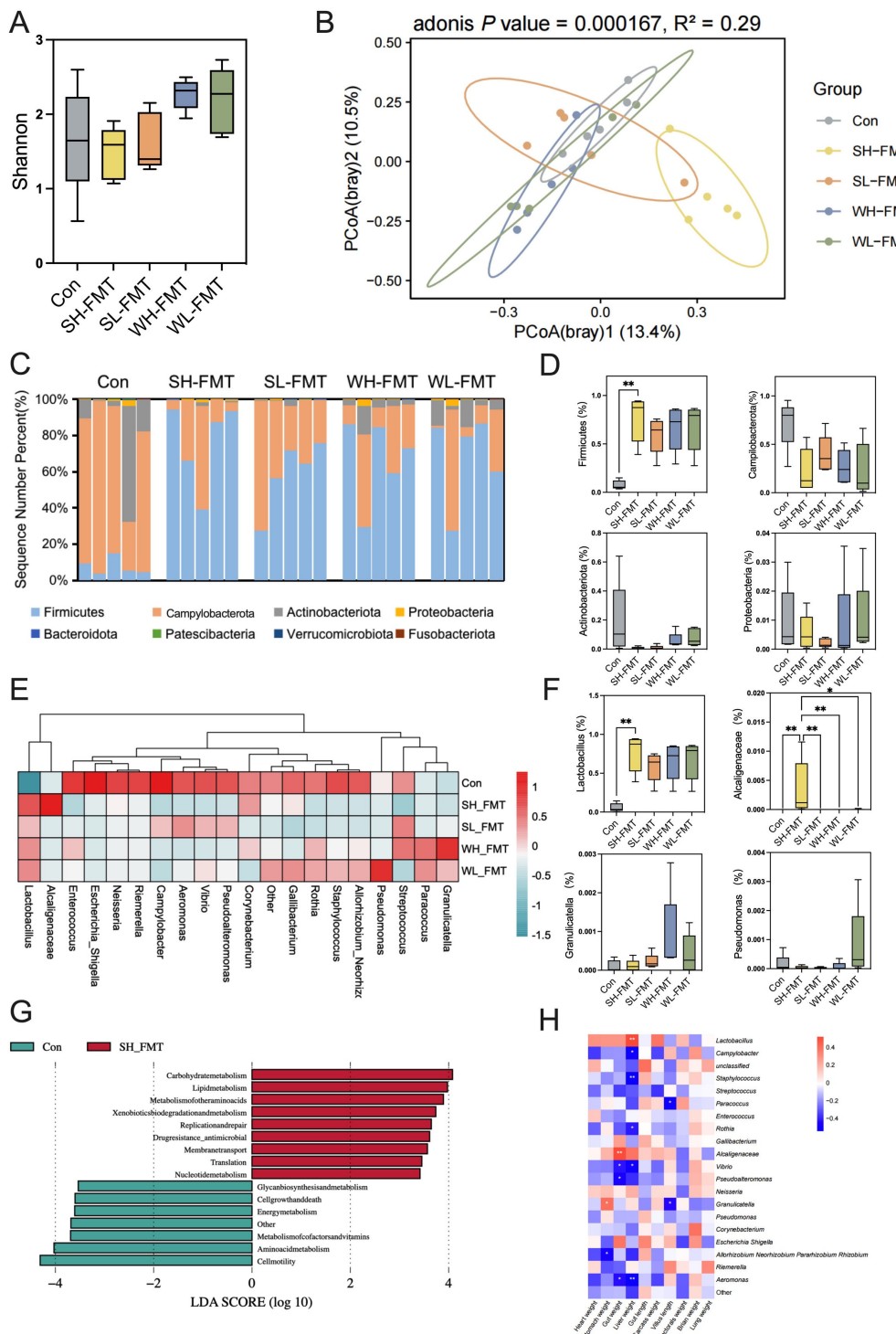

**FIG 3** Comparison of the gut microbiota of zebra finch after FMT. (A) Alpha diversity: Shannon index (Kruskal-Wallis). (B) Principal coordinate analysis (PCoA) plot of the Bray-Curtis distance (PERMANOVA). (C) Result of the phylum level of the gut microbiota in zebra finches. (D) Relative abundance of the specific phylum of zebra finches (Kruskal-Wallis). (E) Cluster heatmap showing the proportions of amplicon sequence variants classified at the genus rank. (F) Relative abundance of specific genus of zebra finches (Kruskal-Wallis). (G) Differential bacterial taxonomy selected by linear discriminant analysis effect size with an LDA > 3.5 in microbiota. (H) Association analysis of microbiome and physiological indexes (Pearson). $*P <$ 0.05, $**P < 0.01$, $***P < 0.001$. $n = 5$.

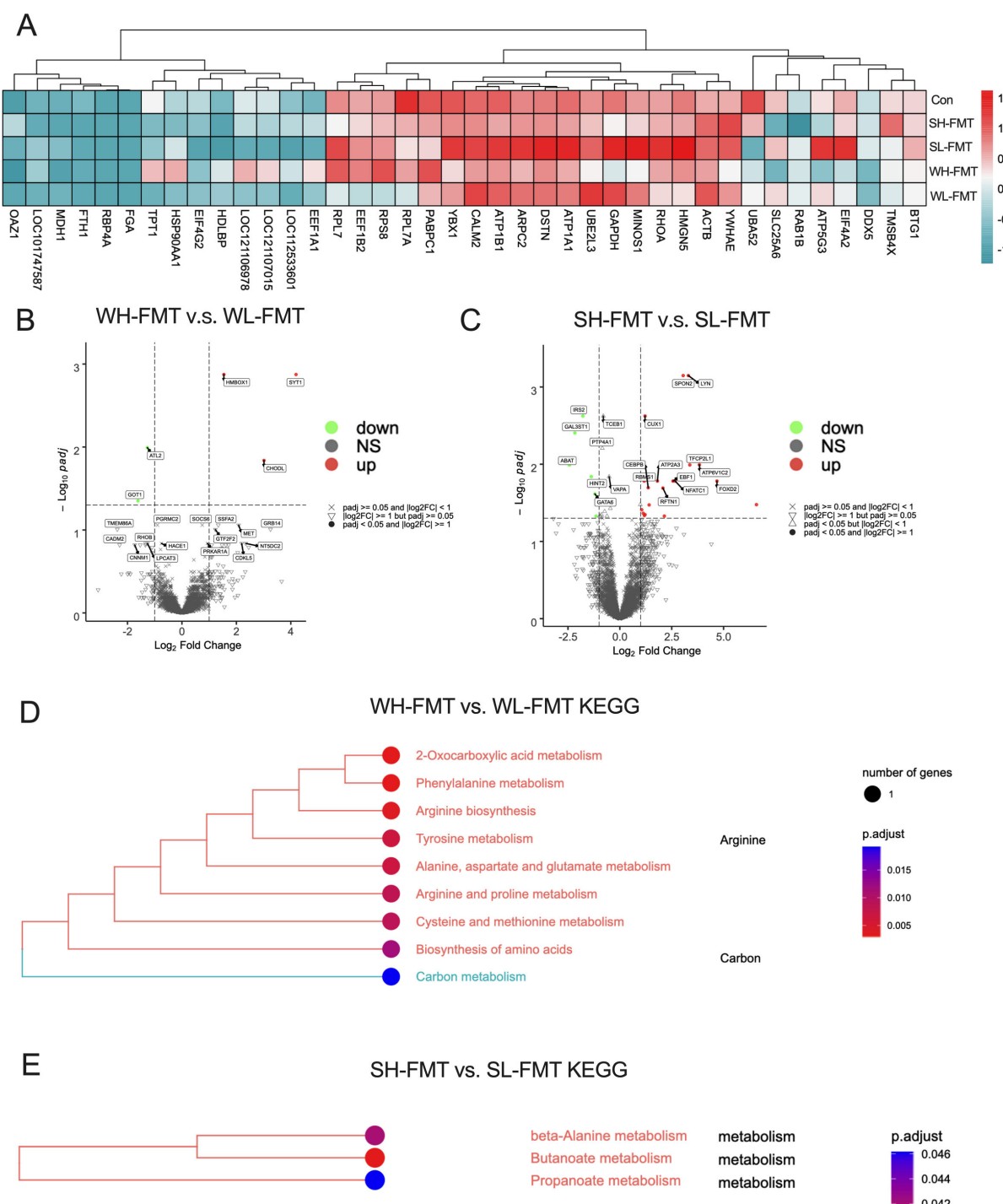

**FIG 4** Comparison of different expression genes of Zebra Finch after FMT. (A) Heatmap showing the abundance of genes ranked high in total abundance in different groups (B) Differentially expressed genes in the gut of zebra finches between WH-FMT and WL-FMT groups. (C) Differentially expressed genes in the gut of zebra finches between SH-FMT and SL-FMT groups. (D) Tree plot of differentially functional pathway enrichment in WH-FMT and WL-FMT groups. (E) Tree plot of differentially functional pathway enrichment in SH-FMT and SL-FMT groups. *n* = 5.

and proline metabolism, and alanine, aspartate, and glutamate metabolism (*P* < 0.05, Fig. 4D). In contrast, the SH-FMT group exhibited enrichment primarily in butanoate metabolism, beta-alanine metabolism, and propanoate metabolism (*P* < 0.05, Fig. 4E).

## DISCUSSION

### Adaptability of the gut microbiota of tree sparrows to high altitude and seasonal changes

The variation in composition and structure of the animal gut microbiota may be an adaptive mechanism to extreme environments (26). In a high-altitude environment, the gut microbiota of birds also undergo adaptive changes. Studies have shown that the gut microbiota of birds in the Qinghai-Tibet Plateau region is closely related to their living environment, with altitude being a major factor influencing their microbial composition (18). Our results show that the gut microbiota composition of tree sparrows is composed of *Firmicutes*, *Proteobacteria*, and *Campylobacter*. The abundance of *Firmicutes* in the gut microbiota of high-altitude tree sparrows is higher than that of low-altitude groups, which can produce more metabolites, such as short-chain fatty acids (SCFAs), for the host to utilize, thereby enhancing the host's metabolism and heat production ability to adapt to extremely cold environments (24, 27). This pattern is consistent with other indigenous species such as yaks and plateau pika (14, 28).

The gut microbiota of tree sparrows exhibits discernible seasonal fluctuations. Specifically, during winter months, the intestinal abundance of *Lactobacillus* and *Clostridium sensu stricto* 1 in high-altitude tree sparrows significantly surpasses that of the summer season. *Lactobacillus* is capable of generating SCFAs, which not only furnish the host with energy (29) but also activate thermogenic signaling pathways under the frigid conditions typical of high-altitude winter environments (24). A multitude of investigations has substantiated the crucial role of *Lactobacillus* in sustaining intestinal equilibrium, including its capacity to produce antimicrobial compounds, such as organic acids and extracellular polysaccharides (EPS). These molecules can effectively impede the colonization of pathogenic strains, such as pathogenic microbes, within the family Enterobacteriaceae (30). Furthermore, *Lactobacillus* has been shown to curtail the production of pro-inflammatory cytokines, markedly diminishing the expression of inflammatory mediators like lipopolysaccharide (LPS) and thus mitigating the onset of intestinal inflammation within the host, thereby upholding a salubrious enteric homeostasis (31). Concurrently, research findings have indicated a significant augmentation in the prevalence of *Lactobacillus* within the gut microbiota in elevated altitude environments (32). Complementing *Lactobacillus*, *Clostridium sensu stricto* 1 contributes to the reduction of enteric inflammation, fortification of the intestinal barrier, and sustenance of intestinal equilibrium (33).

Microbial functional profiling suggests that tree sparrows inhabiting high and low altitudes may exhibit distinct dietary strategies. The gut microbiota of high-altitude tree sparrows in winter is primarily associated with metabolic functions, particularly carbohydrate metabolic pathways, such as fructose and mannose catabolism. This function emphasizes the adaptability of high-altitude tree sparrows to plant seeds and other similar food sources. Conversely, the gut microbiota of low-altitude tree sparrows is more focused on amino acid metabolism, suggestive of a diet richer in proteinaceous foods, including insects and other invertebrates. Prior research has identified a heightened expression of the SLC2A12 gene in high-altitude tree sparrows, which enhances insulin sensitivity and facilitates carbohydrate metabolism (8). This observation indicates a possible correlation between dietary composition and environmental adaptation strategies. In acclimatizing to high-altitude conditions, the gut microbiota may augment the adaptability of tree sparrows to the cold temperatures characteristic of high elevations.

### Effects of high-altitude gut microbiota on host energy balance and metabolic adaptation in birds

Through fecal microbiota transplantation (FMT) from tree sparrows residing at both high and low altitudes during two distinct seasons into zebra finches, we have observed profound alterations in the recipients' gut microbiota composition. Notably, the gut

communities of all zebra finches that underwent FMT differed substantially from the controls. The abundance of *Firmicutes* in the control group was relatively low, but not significant, possibly due to the small sample size. Conversely, the SH-FMT group exhibits an enriched representation of *Firmicutes*, phylum-level bacteria associated with efficient nutrient extraction and short-chain fatty acid production (27), which likely contributes to the augmented body and carcass weights observed in these zebra finches. This finding aligns with previous research indicating that an increased Firmicutes-to-Bacteroidetes ratio correlates with enhanced food resource utilization and may be instrumental in promoting adiposity (34). The high-altitude FMT group displays a concomitant increase in *Lactobacillus*, which are known to confer intestinal homeostasis and aid in carbohydrate metabolism. Correlational analyses further reveal a positive association between *Lactobacillus* abundance and liver weight—a relationship with potential physiological relevance considering the liver's central role in avian thermoregulation and metabolic activity (35). Functional predictions of the microbial community inferred from our data suggest that pathways implicated in energy and lipid metabolism are overrepresented within the high-altitude FMT group. Furthermore, a marked increase in *Alcaliganaceae* has been noted, signifying a potential probiotic benefit through enhanced nutrient absorption for the host (Fig. 3E, 36). Conversely, *Granulicatella*, an opportunistic pathogen (37), is more prevalent in the winter high-altitude FMT (WH-FMT) group, suggesting a possible seasonal immunomodulation in response to high-altitude stressors, mirroring the downregulation of immune-related genes in the gastrointestinal tract of high-altitude tree sparrows.

Birds exhibit fundamental metabolic traits and body temperature regulation mechanisms that are intricately linked to their energy utilization patterns, life history strategies, and evolutionary adaptations. These attributes reflect their profound environmental adaptability and survival capabilities (38). COX, a pivotal enzyme in mitochondrial respiration, is responsible for facilitating the transfer of electrons to molecular oxygen, ultimately leading to the synthesis of water. In the context of seasonal variations, studies have demonstrated that the mitochondrial COX activity in the liver and muscles of winter tree sparrows and white light-vented bulbuls (39) is significantly elevated compared to that observed in summer. This observation aligns with changes in their basal metabolic rate (BMR), suggesting that the COX activity serves as a crucial mechanism underlying the elevation of BMR in small birds (39). Our research further explores the intricate relationship between gut microbiota and thermogenic activity in zebra finches. Notably, the COX activity of zebra finches transplanted with summer microbiota decreased, hinting at a potential link between gut microbiota alterations and reduced thermogenic activity. Conversely, in the winter transplantation group, the expression of avUCP (avian uncoupling protein) protein was upregulated. This finding indicates that the microbial community from high-altitude environments possesses a stimulatory effect on heat production. Moreover, animals residing in high-altitude regions are often subjected to intense selective pressures, resulting in enhanced digestion and absorption capabilities. For smaller individuals, the increased energy demand and food intake become paramount for compensating heat loss under hypothermic and hypoxic conditions characteristic of high-altitude environments (40). Our research results indicate that compared with the low-altitude transplantation group, the winter high-altitude transplantation group has longer intestinal villi, which increases the absorption rate of food and improves the adaptability of birds to harsh environmental conditions.

## FMT indicates the impact of high-altitude microbiota on transcriptional genes

Through transcriptome analysis, we identified some differentially expressed genes related to heat production and energy allocation. The thermogenic activity is activated by the sympathetic nervous system (SNS) through the release of norepinephrine (NE) (41). *Syt1*, as a key regulatory factor for synaptic vesicle release, may be involved in the

process of sympathetic nerve endings releasing NE (42). Skeletal muscles can rapidly produce heat through trembling thermogenesis, and the *Chodl* gene has been shown to be involved in regulating muscle development (43), which may affect thermogenesis efficiency. Muscle secreted muscle factors (such as *irisin*) can induce browning of the white adipose tissue, and *Chodl* may indirectly promote adipose tissue thermogenesis by regulating muscle function. Therefore, the upregulation of *syt1* and *chodl* genes in the WH-FMT group is closely related to increased thermogenesis. With the upregulation of *foxd2*, *ebf1* genes in the SH-FMT group, *foxd2* can respond to photoperiod changes, regulate the hypothalamic pituitary gonadal axis (HPG axis), and initiate seasonal reproductive behavior (such as gonadal development before bird migration) (44). The upregulation of *ebf1* is due to the increased risk of pathogen exposure during the breeding season, and *ebf1* enhances anti-infective ability by enhancing the B cell response. *Cebpb* is upregulated in the SH-FMT group, indicating the promotion of heat-producing gene (UCP1, PGC-1α) expression and the maintenance of body temperature in animals at a constant temperature (45, 46). Despite being in summer, the heat production mechanism of high-altitude birds is still higher than that of the low-altitude group.

In summary, under the stringent environmental selection pressure exerted by the Qinghai-Tibet Plateau, tree sparrows manifest distinctive "gut microbiota-gene" synergistic traits. By modifying their gut microbial communities and gene expression profiles, these birds acclimate to the harsh conditions of high-altitude environments. Our investigations suggest that *Lactobacillus* may be a key player in this adaptation, particularly impactful in modulating sugar metabolism to facilitate heat production and enhance the adaptability of tree sparrows in elevated altitudes. Through microbiota transplantation experiments, we have corroborated the influence of avian microbiomes on energy metabolism and physiological functions. This experiment represents the first instance of its kind conducted on wild birds, setting a precedent for future research endeavors concerning the gut microbiome of wild avian species.

The current experimental design did not delve into the potential impacts of different altitudes on the gut microbiota composition and energy metabolism of tree sparrows. Additionally, due to constraints in field sampling conditions, we are unable to accurately identify the food habits and physiological indicators of wild tree sparrows. This gap in knowledge hinders our understanding of the dietary composition of wild birds and the specific mechanisms through which key gut microbiota influence energy metabolism in avian species. Future research endeavors must aim to address these limitations and provide a more comprehensive analysis of the intricate relationships between gut microbiota, dietary intake, and metabolic processes in birds.

## ACKNOWLEDGMENTS

We would like to express our special gratitude to Jia Jia, Chenxi Jia, Lei Wu, Yifang Zhao, Shiyu Tang, Shangyu Wang (Institute of Zoology, Chinese Academy of Sciences), and Zhiwei Ma (Anhui Normal University) for their assistance in sample collection. We also sincerely thank Chengrui Wang (engineer, Weikemeng Technology Group, China) for assistance with the metabolomics data analysis.

This study was supported by the National Natural Science Foundation of China (32200381 and 32470487), Beijing Natural Science Foundation (5242016), National Key Research and Development Program of China (2022YFC2601601), and the Second Tibetan Plateau Scientific Expedition and Research Program (2019QZKK0304-2).

T.B., G.S., and F.L.: conceived the project and designed the study. M.Z., J.D., H.S., Y.F., X.X., W.L., J.W., and J.L.: field work and lab experiments. D.W. and F.L.: supervision and project management. T.B., G.S., and F.L.: funding acquisition. T.B. and X.X.: data analysis. T.B. and X.X.: writing draft. All authors contributed to the final version of the manuscript.

## AUTHOR AFFILIATIONS

[1]State Key Laboratory of Animal Biodiversity Conservation and Integrated Pest Management, Institute of Zoology, Chinese Academy of Sciences, Beijing, China

[2]School of Grassland Science, Beijing Forestry University, Beijing, China

[3]University of the Chinese Academy of Sciences, Beijing, China

[4]College of Life and Environmental Science, Wenzhou University, Wenzhou, China

[5]School of Life Science, Shandong University, Qingdao, China

## AUTHOR ORCIDs

Tingbei Bo  http://orcid.org/0000-0002-6163-2509
Dehua Wang  http://orcid.org/0000-0002-7322-2371
Fumin Lei  http://orcid.org/0000-0001-9920-8167

## FUNDING

| Funder | Grant(s) | Author(s) |
|---|---|---|
| National Natural Science Foundation of China | 32200381,32470487 | Tingbei Bo |
| Natural Science Foundation of Beijing Municipality | 5242016 | Tingbei Bo |
| National Key Research and Development Program of China | 2022YFC2601601 | Fumin Lei |

## AUTHOR CONTRIBUTIONS

Gang Song, Methodology, Writing – original draft | Mengru Zhang, Formal analysis, Software, Writing – original draft | Xiaoming Xu, Formal analysis, Writing – original draft | Jundong Duan, Formal analysis | Huishang She, Methodology | Yun Fang, Resources | Wenting Li, Software | Jing Wen, Data curation | Jingsong Liu, Validation | Dehua Wang, Supervision, Writing – review and editing | Fumin Lei, Conceptualization, Funding acquisition, Project administration, Resources, Supervision, Writing – review and editing.

## DATA AVAILABILITY

Raw sequence data are deposited in the NCBI Sequence Read Archive under accession BioProject PRJNA1057531.

## ETHICS APPROVAL

All experimental protocols were approved by the Animal Care and Use Committee of the Institute of Zoology at the Chinese Academy of Sciences.

## ADDITIONAL FILES

The following material is available online.

Open Peer Review

**PEER REVIEW HISTORY (review-history.pdf).** An accounting of the reviewer comments and feedback.

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
