## [Reviewer comments · mSystems]

Gut microbiota contribute to high-altitude adaptation in tree sparrows

TINGBEI Bo, Gang Song, mengru zhang, Xiaoming Xu, Jundong Duan, Huishang She, Yun Fang, Wenting Li, Jing Wen, Jingsong Liu, Dehua Wang, and Fumin Lei

Corresponding Author(s): Fumin Lei, CAS Key Laboratory of Zoological Systematics and Evolution, Institute of Zoology, Chinese Academy of Sciences

Review Timeline:

Submission Date:	April 30, 2025
Editorial Decision:	June 16, 2025
Revision Received:	June 26, 2025
Accepted:	July 2, 2025

Editor: Rachel Diner

Reviewer(s): Disclosure of reviewer identity is with reference to reviewer comments included in decision letter(s). The following individuals involved in review of your submission have agreed to reveal their identity: Huan Li (Reviewer #2)

Transaction Report:

DOI: <https://doi.org/10.1128/msystems.00630-25>

Re: mSystems00630-25 (**Mechanism of gut microbiota promoting adaptation of tree sparrows to high-altitude environments**)

Dear Dr. Fumin Lei:

Thank you for the privilege of reviewing your revised work. Based on a full review of your revised manuscript, we have made the decision to accept this paper for publication with minor revisions.

Below you will find my comments, instructions from the mSystems editorial office, and the reviewer comments that should be taken into account for the final submission.

Revision Guidelines

Sincerely,
Rachel Diner
Editor
mSystems

Reviewer #2 (Comments for the Author):

The authors have solved my concern.

Reviewer #4 (Comments for the Author):

Brief summary of the manuscript

This study employed a multidisciplinary approach combining amplicon sequencing, transcriptome analysis, and fecal microbiota transplantation (FMT) to investigate the functional role of gut microbiota in high-altitude tree sparrows across different seasons. The results indicate that the gut microbiota of tree sparrows exhibits seasonal and altitude dependent changes, with an increase in lactobacilli in winter, which may promote heat production to cope with the cold. FMT experiments confirmed that "high-altitude gut microbiota" enhances the expression of heat-related proteins (avUCP) and upregulates heat-related genes *syt1* and *chodl*.

Overall impression of the work

The research presented in this article is very interesting. The FMT experimental design is sound, demonstrating the synergistic relationship between gut microbiota and thermogenic genes, and providing new evidence for avian adaptation to high-altitude habitats. However, some experimental details need to be supplemented to make the data more compelling. Additionally, as a suggestion beyond the manuscript scope: the specific association and mechanism of action between gut microbiota and thermogenic genes remain unclear, it is recommended that future research continue to explore this. Therefore, after appropriate revisions, I recommend this article for publication in mSystems.

Specific comments

1. It is suggested that habitat characteristics of the two groups of tree sparrows, including temperature data across different seasons, be introduced at appropriate points to quantify the temperature differences.
2. Was fecal microbiota transplantation administered via oral gavage? If so, what volume of bacterial suspension or PBS was delivered per administration?
3. Line 264-265: The taxonomic level of the described microbial communities should be specified. In fact, it is not advisable to introduce the characteristics of gut microbiota for all populations so broadly at the genus level; a more detailed discussion of their differences and similarities is warranted. Additionally, the placement of this sentence is incongruous within the text, as it is surrounded by results of gut microbiota characteristics at the phylum level.
4. Line 378: Reference evidence should be provided here to support the statement that "Lactobacillus activate thermogenic signaling pathways under the frigid conditions typical of high-altitude winter environments."
5. Line 478: Attention should be paid here to the proper formatting of Latin names.
6. The description of the small intestinal villus length results should be more comprehensive. Currently, the Results section only addresses the differences in summer groups, whereas the Discussion section focuses on variations across winter groups at different altitudes.
7. Were differences in brown adipose tissue weight observed among the groups in the FMT experiment?
8. The title should be revised as its scope is overly broad. It would be more appropriate to specifically highlight the adaptation of tree sparrows to high-altitude cold climates.
9. The contributions of all authors have not been fully specified in the Contributions section.

Reviewer #5 (Comments for the Author):

Bo et al. has conducted a thorough revision of their manuscript and the improvements in the introduction and the language makes it easy to follow this study. Please go through the results and methods carefully and make sure that all the analyses and measurements you have indicated in the results are explained in the methods section (i.e., Faith_PD measurements are missing in the methods). Below are my comments for the current version of the manuscript.

L 48: indicate that you are talking about "Across the high altitudes of the globe" not just in the Tibetan plateau.

L60: I don't think you meant phylogenetic changes but the changes associated microbial communities. So adjust this accordingly.

L69: Please say what are transgenes and why is it important (very briefly)

L71: This sentence is missing a connecting word after the reference.

L93-95: This need to be bit clearer. Why did you do the faecal transplant and what do you expect from here? Similar gene expression in transplanted individuals and original individuals? Also mention that you used naive birds from a different species to avoid any prior exposure to the conditions. It's also important to defend why you chose a different species for transplants. Try to include some hypotheses related to conducting the transplant experiment here.

L106-110: Can you please be specific about this. Did you euthanised the bird first and then collected the intestinal content? If so, did you dissected the gut out first? I think it is very important to be very clear about your sample protocol for the reproducibility of this study.

L118-119: What about the genetics? Are you not worried that if there are mismatches between gene-microbe interaction in zebra finches and tree-sparrows. May be you can use this as an advantage saying that any changes in the zebra finch genes associated with transplanted gut microbes will represent gut microbe modulation of genes related to high altitudes.

L125: change "sparrow" to "Finches"

L128: This is actually not FMT right? It is the control group where FMT process was mimicked with PBS. So mention that. But then when I looked at the figure 2, it seems like this control group is a fecal transplanted from other zebra finches. Please be clear about what these "controls" actually are.

L132-133: Here it is also important to mention how the donor samples were stored. Were they collected in PBS and kept frozen? Would be good to give this information so others who would like to do similar project can follow the exact protocol :)

L140: Say how exactly you collected this.

L141: Now you say its fecal samples. Be consistent. There is a big difference between intestinal samples and faecal samples as the bird faecal samples are a mix of intestinal, excretory and reproductive system outputs.

L191: Say why you did this as you explain the reason to do the COX activity in the next paragraph

L190, 196, 205: In these paragraphs be specific about which animals you are talking about (zebra finches or house sparrows or both).

L218-223: Good to see these. But did the data met the assumption of homogeneity of variance and normality to run annovas?

Also did you run independent analyses to test the effect of altitude and season? Please say what the models included (e.g., independent variables and dependent variables). Did you test the interaction between these variables?

L232: Not a useful word. Remove.

L233: Say how many ASVs

L240: Indicate the test type and the test statistics such as F and degrees of freedom.

L243: This is the first time you mention Faith PD in the manuscript. Please mention this in the methods and how you calculated this.

L244-245: It's phylogenetic richness. Not evolutionary difference.

L246: here you are talking about Phylogenetic diversity not species diversity right?

L247-248: PCOA is a visualisation method and your actual analysis is PERMANOVA. Make sure to write correct things.

L274: Don't italicise the "sensu stricto 1"

L279: Say Significantly abundant genera instead of higher genera

L310: remove the word "rigorous"

L310-311: What are these quality control measures? I don't see them being specified in the methods section. Make sure to include that in the methods.

L313: Rephrase this sentence. And state what diversity index you are talking about.

L340: Remove the word regorously. This word has no meaning in your context, unless you have done something special to make the differential expression analysis "rigorous". Then include those "special thinkgs you did" in the methods section.

L382: This is a very big family. May be rephrase this to "pathogenic microbes within the family Enterobacteriaceae". Or provide the genera in the family that are know pathogens.

L390: Missing a reference here.

L396-397: Given that you sampled near a city park, can this also represent feeding on human associated food sources?

L409: Remove the phrase "through the bar chart" and make the next section into a new sentence.

L411: change exhibites to exhibits

L444: "in the WH-FMT group" is redundant here.

L478: Italicise *Lactobacillus*

L486: Remove this heading and move this paragraph before the previous paragraph. You don't want to end the ms with insufficiencies.

Remarks to the Author:

Brief summary of the manuscript

This study employed a multidisciplinary approach combining amplicon sequencing, transcriptome analysis, and fecal microbiota transplantation (FMT) to investigate the functional role of gut microbiota in high-altitude tree sparrows across different seasons. The results indicate that the gut microbiota of tree sparrows exhibits seasonal and altitude dependent changes, with an increase in lactobacilli in winter, which may promote heat production to cope with the cold. FMT experiments confirmed that "high-altitude gut microbiota" enhances the expression of heat-related proteins (avUCP) and upregulates heat-related genes *syt1* and *chodl*.

Overall impression of the work

The research presented in this article is very interesting. The FMT experimental design is sound, demonstrating the synergistic relationship between gut microbiota and thermogenic genes, and providing new evidence for avian adaptation to high-altitude habitats. However, some experimental details need to be supplemented to make the data more compelling. Additionally, as a suggestion beyond the manuscript scope: the specific association and mechanism of action between gut microbiota and thermogenic genes remain unclear, it is recommended that future research continue to explore this. Therefore, after appropriate revisions, I recommend this article for publication in *mSystems*.

Specific comments

1. It is suggested that habitat characteristics of the two groups of tree sparrows, including temperature data across different seasons, be introduced at appropriate points to quantify the temperature differences.
2. Was fecal microbiota transplantation administered via oral gavage? If so, what volume of bacterial suspension or PBS was delivered per administration?

3. Line 264-265: The taxonomic level of the described microbial communities should be specified. In fact, it is not advisable to introduce the characteristics of gut microbiota for all populations so broadly at the genus level; a more detailed discussion of their differences and similarities is warranted. Additionally, the placement of this sentence is incongruous within the text, as it is surrounded by results of gut microbiota characteristics at the phylum level.

4. Line 378: Reference evidence should be provided here to support the statement that “Lactobacillus activate thermogenic signaling pathways under the frigid conditions typical of high-altitude winter environments.”

5. Line 478: Attention should be paid here to the proper formatting of Latin names.

6. The description of the small intestinal villus length results should be more comprehensive. Currently, the Results section only addresses the differences in summer groups, whereas the Discussion section focuses on variations across winter groups at different altitudes.

7. Were differences in brown adipose tissue weight observed among the groups in the FMT experiment?

8. The title should be revised as its scope is overly broad. It would be more appropriate to specifically highlight the adaptation of tree sparrows to high-altitude cold climates.

9. The contributions of all authors have not been fully specified in the Contributions section.

Response to Reviewers

We sincerely thank the reviewers for their insightful feedback, which has significantly strengthened our manuscript. Below are point-by-point responses to all comments. Changes are highlighted in the revised manuscript.

Reviewer #4 (Comments for the Author):

Brief summary of the manuscript

This study employed a multidisciplinary approach combining amplicon sequencing, transcriptome analysis, and fecal microbiota transplantation (FMT) to investigate the functional role of gut microbiota in high-altitude tree sparrows across different seasons. The results indicate that the gut microbiota of tree sparrows exhibits seasonal and altitude dependent changes, with an increase in lactobacilli in winter, which may promote heat production to cope with the cold. FMT experiments confirmed that "high-altitude gut microbiota" enhances the expression of heat-related proteins (avUCP) and upregulates heat-related genes *syt1* and *chodl*.

Overall impression of the work

The research presented in this article is very interesting. The FMT experimental design is sound, demonstrating the synergistic relationship between gut microbiota and thermogenic genes, and providing new evidence for avian adaptation to high-altitude habitats. However, some experimental details need to be supplemented to make the data more compelling. Additionally, as a suggestion beyond the manuscript scope: the specific association and mechanism of action between gut microbiota and thermogenic genes remain unclear, it is recommended that future research continue to explore this.

Therefore, after appropriate revisions, I recommend this article for publication in *mSystems*.

Specific comments

1. It is suggested that habitat characteristics of the two groups of tree sparrows, including temperature data across different seasons, be introduced at appropriate points to quantify the temperature differences.

A: Thank you for your good suggestion. We have collected meteorological data on websites: "The temperature difference between the two places is significant. Gahai Zecha National Nature Reserve: The daily average temperature in summer is 8-15 °C, in winter it is -12 to -8 °C, and at night it can reach as low as -30 °C. Beijing Wangjiayuan has an average daily temperature of 25-32 °C in summer, with high temperatures reaching up to 35 °C. In winter, the average daily temperature is -5-5 °C, and the low temperature can reach -10 °C."

2. Was fecal microbiota transplantation administered via oral gavage? If so, what volume of bacterial suspension or PBS was delivered per administration?

A: Thank you. We have provided explanations in the method. "Oral gavage administration, 50uL per day."

3. Line 264-265: The taxonomic level of the described microbial communities should be specified. In fact, it is not advisable to introduce the characteristics of gut microbiota for all populations so broadly at the genus level; a more detailed discussion of their differences and similarities is warranted. Additionally, the placement of this sentence is incongruous within the text, as it is

surrounded by results of gut microbiota characteristics at the phylum level.

A: Yes, the description and location of this sentence are inappropriate, so we have deleted it. The results of LefSe are specific to the genus level, so we will not elaborate on them here.

4. Line 378: Reference evidence should be provided here to support the statement that "Lactobacillus activate thermogenic signaling pathways under the frigid conditions typical of high-altitude winter environments."

A: In our previous research on rodents, we found that "using correlation analyses, we found that at the genus level, Oscillospira and Ruminococcus were correlated with food intake, and Lactobacillus, Oscillospira, Prevotella, Roseburia, and Ruminococcus were correlated with BAT thermogenesis." We have added this reference. Bo TB, Zhang XY, Wen J, Deng K, Qin XW, Wang DH. The microbiota-gut- brain interaction in regulating host metabolic adaptation to cold in male Brandt's voles (*Lasiopodomys brandtii*). ISME J. 2019;13(12):3037-53.

5. Line 478: Attention should be paid here to the proper formatting of Latin names.

A: We italicized it. "*Lactobacillus* "

6. The description of the small intestinal villus length results should be more comprehensive. Currently, the Results section only addresses the differences in summer groups, whereas the Discussion section focuses on variations across winter groups at different altitudes.

A: It was our mistake, I supplemented the results of the small intestine villi. "Histological assessment employing paraffin sectioning and hematoxylin-eosin (HE) staining revealed elongated intestinal villi within the summer transplantation groups; villis in WH-FMT group was longer than WL-FMT group ($P < 0.0001$; Figures 2K,O). "

7. Were differences in brown adipose tissue weight observed among the groups in the FMT experiment?

A: As far as we know, there is still controversy over whether birds have brown adipose tissue, and their heat production mainly relies on the liver and muscles. Therefore, we tested the thermogenic related indicators of the liver and pectoral muscles (Figure 2).

8. The title should be revised as its scope is overly broad. It would be more appropriate to specifically highlight the adaptation of tree sparrows to high-altitude cold climates.

A: Thank you for carefully reviewing and providing valuable suggestions. We understand that you think the original title may be slightly broad in scope and suggest focusing on high-altitude cold climate adaptation. However, after careful discussion within the team, we tend to retain the original title. The reason is that: Our research does include the adaptation mechanisms of tree sparrows in high-altitude cold environments (which will be an important part of the paper), as well as comparisons of different altitude gradients and seasons. Therefore, the original title may provide readers with appropriate research categories.

9. The contributions of all authors have not been fully specified in the Contributions section.

A: We regret such a low-level mistake and have made modifications. "TB, GS, and FL conceived the project and designed the study. MZ, JD, HS, YF, XX, WL, JW, and JL: field work and lab experiments. DW and FL: supervision and project management. TB, GS, and FL: funding acquisition. TB and XX: data analysis. TB and XX: writing draft. All authors contributed to the final version of the manuscript."

Reviewer #5 (Comments for the Author):

Bo et al. has conducted a thorough revision of their manuscript and the improvements in the introduction and the language makes it easy to follow this study. Please go through the results and methods carefully and make sure that all the analyses and measurements you have indicated in the results are explained in the methods section (i.e., Faith_PD measurements are missing in the methods). Below are my comments for the current version of the manuscript.

L 48: indicate that you are talking about "Across the high altitudes of the globe" not just in the Tibetan plateau.

A: Yes, I agree with your point of view. We only use the Qinghai Tibet Plateau as a representative of high-altitude regions around the world.

L60: I don't think you meant phylogenetic changes but the changes associated microbial communities. So adjust this accordingly.

A: We changed the sentence to "Similar results regarding changes associated microbial communities were observed in wild plateau pika...".

L69: Please say what are transgenes and why is it important (very briefly)

A: This is a writing error, we changed the sentence to "transgenes" to "gut microbiota structure."

L71: This sentence is missing a connecting word after the reference.

A: Thank you. We changed to " Previous studies have shown that the stability of bird microbiomes is relatively low (Song et al., 2020; Bodawatta et al. 2021) and the extensibility to environmental and dietary changes is high. "

L93-95: This need to be bit clearer. Why did you do the faecal transplant and what do you expect from here? Similar gene expression in transplanted individuals and original individuals? Also mention that you used naive birds from a different species to avoid any prior exposure to the conditions. It's also important to defend why you chose a different species for transplants. Try to include some hypotheses related to conducting the transplant experiment here.

A: We used the sentence "In order to elucidate the functional impact of the gut microbiome of high-altitude tree sparrows, this study conducted a fecal microbiome transplantation experiment." to show our purpose. We cannot use tree sparrows as transplant objects, we can only select artificially raised model birds - Zebra finches (*Poephila guttata*). They both belong to the Passeriformes order. Described in the method "Zebra finches (*Poephila guttata*) were selected as the recipient organism for microbiota transplantation due to their similar size (approximately 10 cm) and omnivorous diet, primarily composed of seeds, fruits, and insects, conferring robust adaptability."

L106-110: Can you please be specific about this. Did you euthanised the bird first and then collected the intestinal content? If so, did you dissected the gut out first? I think it is very important to be very clear about your sample protocol for the reproducibility of this study.

A: We changed the method. "Euthanizing tree sparrows with anesthetics, dissect and isolate the intestinal tract of the tree sparrow, and place it in a sterile culture dish, and using the intestinal contents taken 1-2cm near the cloaca after execution to put it into a sterile freezer."

L118-119: What about the genetics? Are you not worried that if there are mismatches between gene-microbe interaction in zebra finches and tree-sparrows. May be you can use this as an advantage saying that any changes in the zebra finch genes associated with transplanted gut microbes will represent gut microbe modulation of genes related to high altitudes.

A: Thank you very much for your suggestion. In order to reduce the interference of other factors such as gender, age, and habitat on microbial transplantation, we chose “mice in birds” - Zebra finches (*Poephila guttata*), which better illustrates that any changes in the zebra finch genes associated with transplanted gut microbes will represent gut microbe modulation of genes related to high altitudes. I know there may be genetic influences, but cross species transplantation has also been applied in many studies, such as our previous paper on transplanting elephant feces into mice. (Tingbei Bo; He Liu; Min Liu; Qiyong Liu; Qingduo Li; Yipeng Cong; Yi Luo; Yuqi Wang; Bo Yu; Tianchun Pu; Lu Wang; Zheng Wang; Dehua Wang ; Mechanism of inulin in colic and gut microbiota of captive Asian elephant, *Microbiome*, 2023, 11(1): 1-1)

L125: change "sparrow" to "Finches"

A: We changed it.

L128: This is actually not FMT right? It is the control group where FMT process was mimicked with PBS. So mention that. But then when I looked at the figure 2, it seems like this control group is a fecal transplanted from other zebra finches. Please be clear about what these "controls" actually are.

A: Yes, so we changed to “Con (gavaged with phosphate-buffered saline)”. The control group is designed to simulate the effects of daily gavage and equivalent volume fluid intake on satiety. We changed the figure 2A like this.

L132-133: Here it is also important to mention how the donor samples were stored. Were they collected in PBS and kept frozen? Would be good to give this information so others who would like to do similar project can follow the exact protocol :)

A: Thank you, we added it in the methods. “The donor bacteria were collected from the gut content of wild tree sparrows in Experiment 1, which were collected in PBS and kept frozen, and the content of all tree sparrows were mixed and used in each group.”

L140: Say how exactly you collected this.

A: Euthanizing tree sparrows with anesthetics, dissect and isolate the intestinal tract of the tree sparrow, and place it in a sterile culture dish, and using the intestinal contents taken 1-2cm near the cloaca after execution to put it into a sterile freezer.

L141: Now you say its fecal samples. Be consistent. There is a big difference between intestinal samples and faecal samples as the bird faecal samples are a mix of intestinal, excretory and reproductive system outputs.

A: We are very sorry that our statement is not appropriate. We only took the intestinal contents that come out 1-2cm near the cloaca, similar to feces. It's not a mixture excreted.

L191: Say why you did this as you explain the reason to do the COX activity in the next paragraph

A: In order to understand the status of intestinal villi, we conducted slice experiments.

L190, 196, 205: In these paragraphs be specific about which animals you are talking about (zebra finches or house sparrows or both).

A: Thank you for your suggestion. These laboratory experiments are all aimed at Finches, and we only conducted microbiological testing on tree sparrows. I added this content in the method.

L218-223: Good to see these. But did the data met the assumption of homogeneity of variance and normality to run annovas? Also did you run independent analyses to test the effect of altitude and season? Please say what the models included (e.g., independent variables and dependent variables). Did you test the interaction between these variables?

A: I understand your concerns, and these data conform to the assumptions of homogeneity of variance and normality. We did it before the anova. We conduct a two-way anova analysis, including whether there is an interaction effect of season and altitude. We added it in the results. For example, “a two-way ANOVA test was performed on the bacterial phyla across groups. The results demonstrate that the altitude factor significantly affects the abundance of Firmicutes ($F_{(1, 35)} = 4.789$, $P_{altitude} = 0.035$; $P_{season} = 0.268$; $P_{interaction} = 0.7898$) and Proteobacteria ($P_{altitude} = 0.009$; $P_{season} = 0.273$; $P_{interaction} = 0.3194$),”

ANOVA table	SS (Type III)	DF	MS	F (DFn, DFd)	P value
Interaction	0.008754	1	0.008754	F (1, 35) = 0.07216	P=0.7898
Row Factor	0.5809	1	0.5809	F (1, 35) = 4.789	P=0.0354
Column Factor	0.1536	1	0.1536	F (1, 35) = 1.266	P=0.2682

L232: Not a useful word. Remove.

A: We removed it.

L240: Indicate the test type and the test statistics such as F and degrees of freedom.

A: We added the F and degrees of freedom. For example, “a two-way ANOVA test was performed on the bacterial phyla across groups. The results demonstrate that the altitude factor significantly affects the abundance of Firmicutes ($F_{(1, 35)} = 4.789$, $P_{altitude} = 0.035$; $P_{season} = 0.268$; $P_{interaction} = 0.7898$) and Proteobacteria ($P_{altitude} = 0.009$; $P_{season} = 0.273$; $P_{interaction} = 0.3194$),”

L243: This is the first time you mention Faith PD in the manuscript. Please mention this in the methods and how you calculated this.

A: We have removed the Faith PD result and graph, as well as the description.

L244-245: It's phylogenetic richness. Not evolutionary difference.

A: We have removed the Faith PD result and graph, as well as the description.

L246: here you are talking about Phylogenetic diversity not species diversity right?

A: We have removed the Faith PD result and graph, as well as the description.

L247-248: PCOA is a visualisation method and your actual analysis is PERMANOVA. Make sure to write correct things.

A: Thank you, we changed it. “Principal coordinates analysis (PCoA) based on Bray-Curtis distances visualized distinct clustering patterns among the four groups.”

L274: Don't italicise the "sensu stricto 1"

A: *Clostridium sensu stricto 1*

L279: Say Significantly abundant genera instead of higher genera

L310: remove the word "rigorous"

A: Thank you, we changed them.

L310-311: What are these quality control measures? I don't see them being specified in the methods section. Make sure to include that in the methods.

A: Denoising and Chimera Removal

Sequence Denoising: Amplicon Sequence Variants (ASVs) were resolved using DADA2 v1.20 (Callahan et al., 2016) with parameters:--p-trunc-len-f 270 --p-trunc-len-r 240 --p-max-ee-f 2 --p-max-ee-r 2. Chimera Detection: De novo and reference-based chimeras were removed using the UCHIME algorithm (Edgar et al., 2011) against the SILVA 138 database.

L313: Rephrase this sentence. And state what diversity index you are talking about.

A: It not only considers the richness of species in the community, i.e. the number of different species observed, but also the relative abundance distribution of these species, i.e. evenness. In other words, the Shannon index reflects both the number of species in a community and the degree of uniformity in the distribution of individual numbers among these species.

We changed to "Analysis of the Shannon index revealed a marginally higher gut microbiota alpha diversity in the WH-FMT and WL-FMT groups compared to others, though no statistical significance was observed (Kruskal-Wallis, $P > 0.05$; Figure 3A, Table S4)."

L340: Remove the word regorously. This word has no meaning in your context, unless you have done something special to make the differential expression analysis "rigorous". Then include those "special thinkgs you did" in the methods section.

A: Thank you, we removed it.

L382: This is a very big family. May be rephrase this to "pathogenic microbes within the family Enterobacteriaceae". Or provide the genera in the family that are know pathogens.

A: Thank you, We have made the modifications as per your suggestion.

L390: Missing a reference here.

A: Thank you, we added it. "Li X, Hu S, Yin J, Peng X, King L, Li L, Xu Z, Zhou L, Peng Z, Ze X, Zhang X, Hou Q, Shan Z, Liu L. Effect of synbiotic supplementation on immune parameters and gut microbiota in healthy adults: a double-blind randomized controlled trial. Gut Microbes. 2023 Dec;15(2):2247025. doi: 10.1080/19490976.2023.2247025. PMID: 37614109; PMCID: PMC10453972."

L396-397: Given that you sampled near a city park, can this also represent feeding on human associated food sources?

A: Proximity to urban infrastructure raises the possibility of anthropogenic influences (e.g., consumption of human-provisioned foods or exposure to pollutants). Although direct evidence is lacking in this study, such factors may contribute to the distinct microbiota of tree sparrows. As far as I know, the tree sparrow itself is a species that lives with humans, so it is difficult to avoid interference from human life.

L409: Remove the phrase "through the bar chart" and make the next section into a new sentence.

A: Thank you, we removed it. "The abundance of Firmicutes in the control group was relatively low, but not significant, possibly due to the small sample size. "

L411: change exhibites to exhibits

A: Thank you, we changed it.

L444: "in the WH-FMT group" is redundant here.

A: Thank you, we removed it.

L478: Italicise *Lactobacillus*

A: Thank you, we changed it. *Lactobacillus*

L486: Remove this heading and move this paragraph before the previous paragraph. You don't want to end the ms with insufficiencies.

A: Thank you, you make a really good point, we removed it.

Re: mSystems00630-25R1 (**Gut microbiota contribute to high-altitude adaptation in tree sparrows**)

Dear Dr. Fumin Lei:

I am pleased to inform you that your manuscript "Gut microbiota contribute to high-altitude adaptation in tree sparrows" has been accepted, and I am forwarding it to the ASM production staff for publication. Your paper will first be checked to make sure all elements meet the technical requirements. ASM staff will contact you if anything needs to be revised before copyediting and production can begin. Otherwise, you will be notified when your proofs are ready to be viewed.

Cover Image Submissions: If you would like to submit a potential Cover Image, please email a file and a short legend to mSystems@asmusa.org. Please note that we can only consider images that (i) the authors created or own and (ii) have not been previously published. By submitting, you agree that the image can be used under the same terms as the published article. Image File requirements: TIF/EPS, 7.5 inches wide by 8.25 inches tall (at least 2,250 pixels wide by 2,475 pixels tall), minimum 300 dpi resolution (600 dpi preferred), RGB, and no figure elements, e.g., arrows or panel labels. The legend should be a short description of the image, 1-2 sentences recommended. Please download and use this interactive template in Adobe to ensure that your proposed cover image meets our size requirements (<https://journals.asm.org/pb-assets/pdf-text-excel-files/ASM-Interactive-Sizing-Cover-Template-1715689791.pdf>).

Sincerely,
Rachel Diner
Editor
mSystems